# Inspection Method of Rope Arrangement in the Ultra-Deep Mine Hoist Based on Optical Projection and Machine Vision

**DOI:** 10.3390/s21051769

**Published:** 2021-03-04

**Authors:** Lixiang Shi, Jianping Tan, Shaohua Xue, Jiwei Deng

**Affiliations:** 1School of Mechanical and Electrical Engineering, Central South University, Changsha 410006, China; jptan@csu.edu.cn (J.T.); dengjw@csu.edu.cn (J.D.); 2Nanjing Institute of Electronic Technology, Nanjing 210039, China; shxue163@163.com

**Keywords:** rope arrangement, ultra-deep mine hoist, machine vision, optical projection, boundary extraction, feature extraction, Gaussian filtering, unsupervised clustering

## Abstract

Due to the importance of safety detection of the drum’s rope arrangement in the ultra-deep mine hoist and the current situation whereby the speed, accuracy and robustness of rope routing detection are not up to the requirements, a novel machine-vision-detection method based on the projection of the drum’s edge is designed in this paper. (1) The appropriate position of the point source corresponding to different reels is standardized to obtain better projection images. (2) The corresponding image processing and edge curve detection algorithm are designed according to the characteristics of rope arrangement projection. (3) The Gaussian filtering algorithm is improved to adapt to the situation that the curve contains wavelet peak noise when extracting the eigenvalues of the edge curve. (4) The DBSCAN (density-based spatial clustering of applications with noise) method is used to solve the unsupervised classification problem of eigenvalues of rope arrangement, and the distance threshold is calculated according to the characteristics of this kind of data. Finally, we can judge whether there is a rope arranging fault just through one frame and output the location and number of the fault. The accuracy and robustness of the method are verified both in the laboratory and the ultra-deep mine simulation experimental platform. In addition, the detection speed can reach 300 fps under the premise of stable detection.

## 1. Introduction

Deep resource extraction is an effective method for today’s energy crisis, and it is also an important strategic goal of China [1,2]. Deep resource mining relies on the safe work of mine hoists. For mine hoists where rope hoisting is dominant, the failure of rope arrangement is an important factor affecting the safe and stable operation of hoists [2]. For ultra-deep mines, high speed and heavy load will be common conditions, which generally means that a thicker and longer wire rope is required. This can lead to the width of the rope winding area of the drum and the number of layers of wire rope on the drum increasing. Then, the possibility of rope winding errors is larger, and the risk is greater, making the rope fault detection a problem that must be solved for the ultra-deep mine hoist. However, currently, the detection of the status of the rope arrangement mostly relies on manual inspections or manual guards. The rope tension sensors and displacement transducer installed on the wire rope or drum can tell us some signs of the rope arrangement status, but it is not direct enough, easy to be disturbed, with low accuracy and is unable to find the rope arrangement fault in time. Therefore, a sensor or a detection method for accurate full-time automatic detection of the rope arrangement state is very necessary.

At present, machine vision is increasingly being used for the testing and calculation of specific targets in large-scale projects, and has achieved good results. Such as the calculation and measurement of large-scale structural parts [3], the calculation of the vibration mode of cable-stayed bridges [4,5], and the deformation monitoring [6,7] and geometric shape detection [8] of bridge structures. Using machine vision to detect the status of the wire rope is becoming an effective method.

In the use of machine vision to detect the position of the wire rope, Chen et al. [9] use a customer grade camcorder and an ordinary tripod to detect the frequency of steel wire ropes on a cable-stayed bridge, achieving the same order of accuracy for cable frequency identification as that of high-resolution velocimeters. Winkler et al. [10] use digital image correlation to obtain measurement of local deformations in steel mono strands. Yao et al. proposed a non-contact video-based measurement for transverse displacements of hoisting catenaries in mine hoist using mean shift tracking [11,12] and digital image processing techniques [13]. Wu et al. [14] proposed a non-contact and unmarked machine vision measurement method for measuring the transverse vibration displacement of hoisting vertical ropes. In the hoisting cape area, because the video of the rope detection is shot with the sky as the background, the target and the background are very distinguishable, so the recognition effect is obvious, and the background of the wire rope to be detected in the reel area is the same wire rope (such as Figure 1a) and the target and the background are very similar, thus the same recognition method will be much worse.

Table 1 shows the current existing detection methods for rope arrangement detection. Wu et al. [15] used the method of adaptive gray threshold segmentation on the same horizontal line to achieve the distinction between the target wire rope and the background, but this method is based on the difference in the reflective performance of the wire rope and the rope groove. For the oily wire rope, the tightly wound wire rope and multi-layer steel wire rope on the drum are not applicable. Xue et al. [16] proposed the method of using template matching and limiting the size of the target selection area making the target and the background better to distinguish, and realized the video tracking of the position of the wire rope. Other video tracking methods, like KCFs (kernelized correlation filters) [17] and Staple [18], can be used for rope arrangement detection. However, because the target selection area is limited, the algorithm is easy to lose the target when the rope winding speed is fast or the skipping span is large. At the same time, when shooting a wide reel, the image will inevitably be distorted (as shown in the Figure 1a). In addition, because the target is deformed greatly, part of the texture information is lost and cannot be retrieved through distortion correction, thus the video tracking of the position of the wire rope is very difficult. To sum up, the difficulties of rope detection in ultra-deep mine hoist based on machine vision are summarized as follows:The target is extremely similar to the background;The wider the scroll, the larger the scroll deformation in the projection image;The detection speed and robustness should meet the requirements of high speed.

Therefore, this study aims at detecting and diagnosing the problems of rope arranging faults in ultra-deep mines with wide drums and high speeds, and proposes a new method for arranging rope detection based on machine vision and optical projection. In this study, edge projection image is used to detect the state of rope arrangement to make the rope separated from its complicated and similar background. Using approximate parallel light irradiation and quantitative calculating, the installation position of the point light source required by the drum, with different widths to solve the problem of distortion of the wide reel, is found. For higher detection speed and more accurate detection results, unlike other video tracking methods that require a comparison of several frames, we get the rope arrangement at that moment with just the current single frame and consider the detection results of current frames to judge the rope arrangement fault. In both the laboratory and the simulation test bed, the purpose of error-free identification of the rope fault is achieved, and the processing speed exceeds 150 fps (for narrow reels, it can reach 300 fps or more).

The flow of the novel method is shown in Figure 1. First, illuminate the edge of the reel rope with parallel light, place a pure white background board at its projection position, then use an industrial camera to shoot the projection of the rope on the background board, process the projected image, and extract the characteristic value corresponding to each wire rope. Finally, the unsupervised clustering method is used to classify and get the rope status detection result.

The arrangement of this paper is as follows: At present, there are few introductions of the reel-rope projection model, which has a greater impact on the setting of the relevant parameters of this study. Therefore, this article will describe (in Section 2) how to choose the drum-rope projection model and the installation position of the point light source, (in Section 3) how to quickly obtain the boundary curve from the binarized image and then extract the feature value to ensure that each feature value corresponds to a wire rope or a rope groove; and (in Section 4) how to use the unsupervised clustering method for such features’ values for effective and robust classification. And in Section 5, the application effects in the model machine and the real mine are described.

The meanings of abbreviations and symbols used in this paper are shown in Table 2.

## 2. How to Get the Required Projection

In order to obtain a clear cross-sectional projection of the multiple loops of steel wire rope on a drum, it is necessary to project the light on the plane composed of the light source and the axis of the drum, and the projection light should be approximately parallel to the projection of all steel wire ropes on this plane. The wire rope loops on the reel are approximately parallel to each other, so, in theory, parallel light needs to be used to get the projection of the rope. In reality, we can use a point light source at long distances to produce approximate parallel light. From this point of view, the farther the point light source is installed the better; however, considering the difficulty of long-distance installation, site limitations, susceptibility to external interference and projection light intensity requirements, the actual installation height is limited. Therefore, finding a suitable installation height is very important. The flowchart of this section is shown in Figure 2.

### 2.1. Projection Model of the Drum’s Rope Arrangement

Because the diameter of the drum is much larger than that of the wire rope, the wire rope wound on the drum can be regarded as a ring arranged along the axial array of the drum when calculating the projection of the rope arrangement of the drum. The projection of a single torus is the shape formed when the central circle of the torus extends the radius of the torus cross-section both inside and outside.

**Theorem** **1.**
*The shortest distance from any point on the torus projection to the central circle projection of the torus is the radius of the torus section circle.*


**Proof** **of** **Theorem** **1.**A sphere with the same radius as the cross-section circle of the torus, whose center moves around the central circle of the torus, passes through the space that is the whole space of the torus. Therefore, the projection of a torus is a collection of projections of a sphere centered on a point on the central circle of the torus. Because the projection of a sphere at any projection angle is a circle, and the projection of a circle is an ellipse, so the projection of a torus is the collection of all circles with the ellipse projection of the central circle of the torus as the center of the circle, and the radius of the cross-section circle of the torus as the radius. □

Namely:(1)$$\begin{array}{c} \forall \;\left(x,y\right)\in S_{C}\frac{x^{2}}{{\left(R\times \cos\left(\alpha \right)\right)}^{2}}+\frac{y^{2}}{R^{2}}=1\\ \exists \;\left(x_{t},y_{t}\right)\in S_{Y}\min\left(\mathrm{dist}\left(\left(x,y\right),\left(x_{t},y_{t}\right)\right)\right)=r \end{array}$$
where $S_{C}\;$ represents the central circle projection of the torus, $S_{Y}$ represents the external projection of the ring,  R is the radius of the central circle of the torus, r is the radius of the section circle of torus, and α represents the angle between the projection light and the plane of the central circle of the torus (the same below).

#### 2.1.1. Projection of the Single Coil Wire Rope

The projection of the single coil wire rope is considered. Figure 3 below shows the projection of the torus corresponding to a steel wire rope when the angle α, between the projection light and the plane where the central circle of the ring is located, is 90°, 60°, 30° and 0°. The four small circles in the four small figures correspond to the positions of 90°, 60°, 30° and 0° on the dividing circle of the torus, respectively. In general, the projection of the torus produced by the usually installed point light source is between α = 30° (Figure 3c) and α = 0° (Figure 3d).

#### 2.1.2. Projection of Multi Coil Wire Ropes

When multi circle wire ropes are projected, there will be projection interference between two adjacent wires, which will affect the follow-up classification effect and reduce the reliability of detection. When the angle α between the projection light and the axis of the torus increase, the resolution of a single wire rope in the projection will reduce. An index is needed to represent the relationship between the resolution of a single wire rope and the angle α. Therefore, the ratio η between the height difference (dy) of the projection intersection point and the highest projection point and the section radius (r) of the wire rope is used as the projection quality index (as shown in the Figure 4). The larger η, the higher the resolution of single wire rope in projection.

In the Figure 4, ‘A’ is the intersection point of the projection outer rings of two torus. Namely:(2)$$\begin{array}{c} \left(x_{1},y_{1}\right)\in S_{C1},\;\left(x_{2},y_{2}\right)\in S_{C2},\;\\ \frac{x_{1}{}^{2}}{{\left(R\times \cos\left(\alpha \right)\right)}^{2}}+\frac{y_{1}{}^{2}}{R^{2}}=1\\ \frac{{\left(x_{2}-2\times r\right)}^{2}}{{\left(R\times \cos\left(\alpha \right)\right)}^{2}}+\frac{y_{2}{}^{2}}{R^{2}}=1\\ \exists \;\mathrm{Intersection}\;A\;\left(x_{t},y_{t}>0\right)\;\;\;\;\;\;\;\;\;s.t.\;\;\;\left(x_{t},y_{t}\right)\in S_{C1},\;\left(x_{t},y_{t}\right)\in S_{C2} \end{array}$$
where R is the radius of the drum and r is the radius of the wire rope. $\left(x_{1},y_{1}\right)$ and $\left(x_{2},y_{2}\right)$ are points on the projection of the outer rings of the two torus, respectively.

Then calculate dy and η as follows:(3)$$\begin{array}{c} \min\left(\;\mathrm{dist}\left(\left(x_{1},y_{1}\right),\left(x_{t},y_{t}\right)\right)\right)=r\\ \min\left(\mathrm{dist}\left(\left(x_{2},y_{2}\right),\left(x_{t},y_{t}\right)\right)\right)=r\\ \mathrm{dy}=R+r-y_{t}\\ \eta =\frac{\mathrm{dy}}{r}\times 100\% \end{array}$$

Set the ratio of drum diameter to wire rope diameter as $N_{d}$:(4)$$N_{d}=\frac{R}{r}$$

The relationship between η and α and $N_{d}$ is calculated, and the results are shown in the Figure 5. It is found that when η is less than 50%, the projection of wire rope is easily affected by oil stain, irregular bulge and noise in the projection image, which makes the robustness of the subsequent classification effect not high. η is related to α and $N_{d}$, and η decreases monotonically relative to α. η is related to the ratio $N_{d}$ of drum diameter to wire rope diameter. The larger $N_{d}$ is, the smaller η is.

### 2.2. Determine the Installation Position of Point Light Source

Considering that the horizontal installation of the point light source is easily disturbed by the environment, it is generally selected to install the point light source in the vertical direction of the drum center. According to the relationship between η, α and $N_{d}$ determined in Section 2.1, the installation height $H_{light}$ of the point light source is calculated according to the Equation (5):(5)$$H_{light}\geq \frac{L_{g}}{2\times \tan\left(\alpha \right)}$$
where $L_{g}$ is the width of the drum. The ratio of the installation height of the light source and the width of the drum is μ:(6)$$\mu =\frac{H_{light}}{L_{g}}=\frac{1}{2\times \tan\left(\alpha \right)}$$

For commonly used $N_{d}$ and α, the values of μ are shown in Table 3.

The ratio of the drum width to the wire rope diameter $N_{g}$ is also the maximum number of winding coils of the drum:(7)$$N_{g}=\frac{L_{g}}{2\times r}$$

For the mine hoist, the load capacity determines the radius of the wire rope R, the radius of the wire rope limits the radius of the drum, and the lifting height determines the length of the rope, which also determines $N_{g}$. $N_{g}$ and the radius of the wire rope determine $L_{g}$. According to $N_{d}$ and the selected η, μ can be determined and the suitable range of $H_{light}$ can be determined.

The larger the installed height $H_{light}$ of the point light source, the more space and power of the point light source are needed. It is also necessary to select $H_{light}$ according to the space limitation of the site and the difficulty of light source arrangement. Under comprehensive consideration, the installation height of the point light source is selected according to the standard of 70% of η.

## 3. Boundary Extraction and Feature Extraction of Rope Arrangement Projection

The projection image of drum rope arrangement (partly shown in Figure 6a) has its unique characteristics:There is a big difference between the shadow area and the illumination area;The gray value of shadow area in the image is stable and average, and there is no light noise;The upper limit of the gray value of the illumination area of the light source in the image is affected by the change of the light intensity of the external environment, and the lower limit is affected by the light intensity of the point light source here;Due to the influence of dust and oil on the wire rope, the project of the wire rope may be irregular and the gap between each other becomes unclear;Due to the influence of dust and oil on the background plate, there may be discrete dark spots in the light source area of the image (as shown in Figure 6c).

It is necessary to design the boundary contour acquisition method according to these characteristics to obtain smooth boundary contour curve without repeated region, so as to ensure the accuracy of the subsequent extracted feature values and one-to-one correspondence with each wire rope or rope groove. The flowchart of this part can refer to Figure 1.

### 3.1. Boundary Extraction

There are many methods to obtain contour curve, among which the typical ones are Roberts [19,20], Sobel [20,21], Prewitt [20,22], LoG (Laplacian of Gaussian) and Canny [23,24] operators. The LoG operator can always detect more details than the Canny operator in the same scale, while the Canny operator is not easy to be disturbed by noise.

Due to feature 3 of the rope arranging projection image, the light intensity of the image in the whole-day monitoring video will change greatly, and the detection result of the Canny operator is more sensitive to the double threshold parameters of the edge. If the Canny operator is used directly to the original image, it is easy to produce multi-edge or edge interruption problems (Figure 6e), which will affect the rope arrangement detection result. Using gray threshold segmentation to get the binary image first, and then using Canny or LoG operator-based edge detection methods can effectively suppress this problem, but there is still a problem of multiple values in the same column (Figure 6f). In addition, due to feature 5 of the rope arrangement projection image, when there is an oil stain or other foreign matters in the background plate (Figure 6c,d), the binary image will inevitably have the problem of multiple regions.

Therefore, limiting the light intensity of the external environment and increasing the light intensity of the point light source appropriately can obtain the gray image with stable gray value. The ideal binary image can be obtained by gray threshold segmentation, and this threshold is applicable to the whole video cycle.

Then according to feature 2 of the rope arrangement projection image, the edge curve is calculated as follows:(8)$$t_{i}=\mathrm{find}\left(\mathrm{img}\left(:,i\right)==1\right),S_{i}=t_{i}\left(1\right)\;,\;i\in \left(1,n\right)$$
where S is the edge curve and n is the total number of image columns, img() is a two-dimensional array which represents the ROI of the frames, and find(equation) is a function of find elements that match the equation. The image of the contour curve is shown in Figure 6g.

### 3.2. Feature Extraction

In order to extract the characteristic value of the above-mentioned contour curve S, the uniqueness of the eigenvalue should correspond to each wire rope or rope groove. The peak value or valley value of the curve can meet this condition, so the peak value of the contour curve is selected as the characteristic value. However, due to feature 4 of the rope ranging projection image mentioned above, the oil and dust on the surface of the wire rope will make the curve S not smooth (as shown in Figure 6g, there will be fluctuations near the peak value or valley value of the original curve, showing the phenomenon of multi peak or multi valley value), which will affect the peak value feature extraction. Therefore, it is necessary to carry out low-pass filtering on curve S with the minimum influence on the peak value of the data curve. Mean filtering, middle finger filtering and Gaussian filtering are common low-pass filtering methods. Qiao et al. [25] used Gaussian filtering to smooth the original data in the first step. Compared with mean filter and median filter, Gaussian filter can obtain a smooth curve and retain the peak value better. Therefore, this paper used the Gaussian filter to smooth the contour curve S.

The Gaussian filtering method is shown in Equation (9), in which the variance σ is the key parameter. σ determines the size of the filtering window and the weight of the data in the window, and is often designed as a fixed value. In view of the fact that there may be multiple wavelet peaks at one large peak of curve S, in order to obtain a smooth curve S_1_, the fixed value of σ will be larger for the whole curve, so that all peaks of the S_1_ curve obtained will be reduced. Therefore, an adaptive algorithm is designed to calculate σ. In each filtering window, the algorithm adjusts σ according to the situation that the data fluctuates violently in this window, so that where there are multiple wavelet peaks, σ takes a larger value to ensure that there is only one peak value, while in other places, σ takes a smaller value to retain a larger peak value.
(9)$$\begin{array}{c} G\left(x\right)=\frac{1}{\sqrt{2\pi }\sigma }e^{\frac{{\left(x-d\right)}^{2}}{2\sigma^{2}}},x\in \left[1,\;2,\ldots ,\;2d+1\right]\\ {\mathrm{SF}}_{i}=[S_{i-d},S_{i-d+1},\ldots ,S_{i+d}]\times \frac{G^{\prime }}{\mathrm{sum}\left(G\right)}; \end{array}$$

First, the fluctuation of curve s is calculated as follows:(10)$$\begin{array}{c} \mathrm{DS}=\mathrm{sign}\left(\mathrm{diff}\left(S\right)\right)\\ \mathrm{DDS}=\mathrm{abs}\left(\mathrm{diff}\left(\mathrm{DS}\right)\right) \end{array}$$
where diff() is a function of differential algorithm, sign() is a function to extract symbols and abs() is a function to calculate the absolute value. As the curve S wave is a group of discrete points, it is linearized by the Gaussian function:(11)$${\mathrm{GDDS}}_{i}=[{\mathrm{DDS}}_{i-d},{\mathrm{DDS}}_{i-d+1},\;\ldots ,\;{\mathrm{DDS}}_{i+d}]\times \frac{G^{\prime }}{\mathrm{sum}\left(G\right)}$$
where sum() is a function to calculate the sum.

Then, σ is calculated according to the fluctuation of curve S:(12)$$\sigma_{i}=a\times e^{{\mathrm{GDDS}}_{i}}+b$$

Generally, set a to 1 and b to 0. Figure 7 is the result of the first frame of video 4 (shown in Section 5.2) using our method. The blue line is the fluctuation value of the calculated curve S where there are small peaks, the fluctuation value is 2, where there are normal peaks and troughs, the fluctuation value is 1, and the rest is 0.

Then use Equation (13) to calculate the filtered data:(13)$$\begin{array}{c} G_{i}\left(x\right)=\frac{1}{\sqrt{2\pi }\sigma_{i}}e^{\frac{{\left(x-d\right)}^{2}}{2\sigma_{i}{}^{2}}},x\in \left[1,\;2,\ldots ,\;2d+1\right]\\ {\mathrm{SF}}_{i}=[S_{i-d},S_{i-d+1},\;\ldots ,\;S_{i+d}]\times \frac{{G_{i}}^{\prime }}{\mathrm{sum}\left(G_{i}\right)} \end{array}$$

The peak value and valley value are calculated using the method of finding the peak value. The valley value is greatly affected by the projection quality, so the peak value is set as the characteristic value of curve S that is needed.

The filter window size of the three methods is set to the same as d. Since the half period of wavelet peak is about 2–4 pixels, in order to ensure the robustness of filtering effect, d should not be less than 4. For different window sizes, the retention of peak valley difference by the three filtering methods is shown in Figure 8 and Figure 9. Our method can keep the ratio of peak to valley difference higher than 95% for different window sizes. In order to ensure the robustness of the filtering effect in the whole video, we set d as 6 (the mean period of wavelet peak). The average ratio of the peak to valley difference in frame 1 to frame 1000 of video 4 using our method is 95.4%, while Gaussian smoothing is 92.8% and mean smoothing is 89.2%. Meanwhile, the average retention rate of the peak valley difference at the projection of the wire rope here is 96.5%, and the average retention rate of the peak valley difference at the projection of the rope groove is 90.5%. Figure 9 shows the peak values (eigenvalues) of frame 1 of video 4 with the novel adaptive Gaussian smoothing.

## 4. Unsupervised Clustering

The eigenvalues obtained above grows into strip curve distribution. Figure 10 [26] shows the classification result of eight common clustering classification methods on five selected linear datasets [26]. Among the common clustering algorithms, spectral clustering [27], DBSCAN (density-based spatial clustering of applications with noise) [28], OPTICS [29] (ordering points to identify the clustering structure) and Gaussian mixture [30] are better for this kind of data classification [26]. However, OPTICS has high time complexity and is time-consuming [31], so it is not suitable. Spectral clustering and Gaussian mixture need to take the number of categories as the initial condition, which is not suitable for the case of an unknown number of categories in this study. Whereas, DBSCAN does not need to know the number of categories in advance, clusters dense data sets of any shape, is good at finding outliers, needs just two parameters and has relatively low complexity and fast operation. Therefore, this paper uses DBSCAN to cluster the eigenvalue data.

### 4.1. DBSCAN

DBSCAN is one of the unsupervised density clustering methods. The principle [28,32] is as follows:

Consider a set of N data points $Z\equiv {\left\{z_{n}\right\}}_{n=1}^{N}$.

We start by defining the ε-neighborhood of point $z_{n}$ as follows:(14)$$N_{\varepsilon }\left(z_{n}\right)=\left\{z\in Z|\mathrm{dist}\left(z,z_{n}\right)<\varepsilon \right\}$$

$N_{\varepsilon }\left(z_{n}\right)$ are the data points that are at a distance smaller than ε from $z_{n}$. We consider dist() to be the Euclidean metric (which yields spherical neighborhoods), but other metrics may be better suited depending on the specific data. $N_{\varepsilon }\left(z_{n}\right)$ can be seen as a crude estimate of local density. $z_{n}$ is considered to be a core-point if at least **minPts** (a free parameter of the algorithm that sets the scale of the size of the smallest cluster one should expect.) are in its ε-neighborhood. Finally, a point $z_{i}$ is said to be density-reachable if it is in the ε-neighborhood of a core-point. From these definitions, the algorithm can be simply formulated:
Until all points in Z have been visited; do(1)Pick a point $z_{i}$ that has not been visited;(2)Mark $z_{i}$ as a visited point;(3)If $z_{i}$ is a core point; then:(1)Find the set C of all points that are density reachable from $z_{i}$.(2)C now forms a cluster. Mark all points within that cluster as being visited.Return the cluster assignments $C_{1}$, …, $C_{k}$ with k the number of clusters. Points that have not been assigned to a cluster are considered noise or outliers.

Note that DBSCAN does not require the user to specify the number of clusters but only ε and **minPts**. While it is common to heuristically fix these parameters, methods such as cross-validation can be used for their determination. Finally, we note that DBSCAN is very efficient since efficient implementations have a computational cost of $O\left(n\;\log\left(n\right)\right)$.

### 4.2. Automatic Optimization of DBSCAN Distance Setting

For linear data such as rope arrangement projection, the parameter **minPts** is able to be determined. The key is how to determine the parameter ε. From the classification results of Figure 10d,e datasets, the eigenvalues of rope and rope groove can be distinguished correctly using Euclidean metric to calculation the parameter ε, but the eigenvalues of wrong rope arrangement cannot be classified correctly.

Consider the rope arrangement process. When the rope arrangement is normal and the projection angle α is approximately 0°, the theoretical rope arrangement projection is shown in the Figure 11. The red points (P_1_, P_2_, …, P_n_) are the eigenvalues (peak values) extracted above, and the distance between two adjacent points is d_1_ to d_n−1_ in order. Figure 12 shows the error chart of their distance value, d_i_ is the longest, which can help us divide the data before and after it into two categories.

When the rope arrangement is abnormal, there are two kinds of errors (rope skipping error and rope clamping error). When the rope skipping error occurs, the current winding wire rope leaves the current rope groove and goes to the later rope groove, which may span one or more rope grooves. The projection of spanning one rope groove is shown in Figure 13, and the error diagram of the corresponding characteristic point spacing is shown in Figure 14. The skip angle (as β  in Figure 13 and Figure 14 below) is 180°  in total.

When the rope clamping error occurs, the current winding steel wire rope leaves the current rope groove and climbs above the previous rope, which often jumps one rope or more. The projection is shown in Figure 15, and the error diagram of the corresponding characteristic point spacing is shown in Figure 16. The skip angle (as β in Figure 15 and Figure 16 below) is 120°  in total. In addition, while β is near 60°, the peak near the skipping rope will be covered or exposed, making the total numbers of eigenvalues (red points in Figure 15b) decrease by 1 or not.

A good distance threshold needs to enable DBSCAN to correctly classify the eigenvalues in the whole process of the rope arrangement faults, that is, to ensure that the distance calculation value of the wrong rope arrangement sequence is sufficiently distinguishable from the distance calculation value of the normal rope arrangement sequence. According to Figure 12, Figure 14 and Figure 16, the threshold we need here should be the bilateral threshold. Thus, if the distance between two eigenvalues satisfies the following formula, the latter eigenvalue is the ε-neighborhood of the eigenvalue ahead.
(15)$$\left\{\begin{array}{c} a\times 2r-e_{1}<d_{i}<a\times 2r+e_{1}\\ b\times 2r-e_{2}<d_{i}<b\times 2r+e_{2} \end{array}\right.$$

## 5. Application Effect

In order to test the effectiveness and real-time of this method, we carried out experiments in the laboratory environment and the ultra-deep mine simulation experimental platform, which is close to the ultra-deep mine working condition. According to the above, the parameter η was set to 70% both in the laboratory and ultra-deep well simulation test bench, the projection distance was calculated, and the point light source was installed accordingly.

### 5.1. Application Effect in the Laboratory

The drum width of the laboratory is large, and the installation distance of the point light source was calculated to be 10 m. The hardware layout is shown in the Figure 17. And the specific parameters are shown in the Table 4. The LED light source was installed 10 m above the drum with a white background plate under the drum, and an industrial camera was installed on the side with the field of view facing the projection area on the background plate.

By giving a transverse force to the wire rope winding the drum, the rope arrangement fault was artificially created. We took the projection of the rope in the whole process. The effectiveness and real-time performance of the detection method under different conditions were verified by setting a different number of rope arrangement faults. By setting the distance between the industrial camera and the projection, the pixel size of the ROI area was changed to find the appropriate resolution of the ROI area.

A total of four groups of videos were taken, among which videos 1 and 2 were not set with the rope laying fault, videos 3 and 4 were set with one and two rope laying faults, respectively. Video 1 was shot far away from the drum, video 2 was shot nearby, videos 3 and 4 were shot a moderate distance away from the drum. Four image datasets are obtained by clipping the ROI regions in video images. The predicted frame numbers and time consumption of the rope laying fault using our method are shown in Table 5. Among them, the prediction results before and after the fault in video 4 are shown in Figure 18. The detection time is the total time consumed to detect all the frames in the video and the detection speed is calculated as mean frames detected per second.

### 5.2. Application Effect in the Ultra-Deep Well Simulation Test Bench

The ultra-deep well simulation test bench is a 10:1 model of a hoist with a lifting height of 1500 m, a lifting speed of 18 m/s and a payload of 50 t. The drum width of the laboratory is small, and the installation distance of the point light source is calculated to be 3.5 m. The hardware layout is shown in the Figure 19 and the specific parameters are shown in the Table 6. The LED light source was installed 3.5 m above the drum with a white background plate under the drum and on a red plate, and an industrial camera was installed on the side with the field of view facing the projection area. The smallest graph of Figure 19 shows the actual projection image.

A total of two groups of videos were taken, of which video 1 did not set the rope fault and video 2 set one rope fault. Two image datasets are obtained by clipping the ROI region in the video image. The results are shown in Table 7.

## 6. Discussion and Conclusions

In this study, through the research of the drum rope projection model, the appropriate point-light installation positions corresponding to different drums are calculated quantitatively. The corresponding image processing and edge curve detection algorithms are designed according to the characteristics of rope projection. The Gaussian filtering algorithm is improved to adapt to the situation that the curve contains wavelet noise when extracting the eigenvalues, and the results show that this method can retain the peak to the maximum extent. Then, the optimization of the distance calculation function of the DBSCAN method is proposed in order to get a clustering classification method that is suitable for the linear data set and an unknown number of classifications.

The results of the experiments in the laboratory and the ultra-deep mine simulation experimental platform show that:For the different widths of the drum, this method can always detect the rope arrangement state stably and at high speed;If the ROI area width of the projection image exceeds 50 pixels, the subsequent correct detection can be ensured;The detection speed is related to the image size. When the image width is set to 70 pixels, the detection speed of the reel with 14 turns exceeds 300 fps. However, when the image width is set to 50 pixels, the detection speed of the reel with 95 turns exceeds 200 fps;The detection sensitivity of this method is higher than that of human eye observation, and it can detect the rope arranging fault earlier.

Compared with other methods and manual inspection, the innovation of this study lies in that:This study reduces the dimension of the difficulties of the detection of rope arrangement by projection;This study replaces the problem of inevitable image distortion and overlap distortion in wide-roll shooting with the installation space requirement of the light source, which can be easily solved;This study can judge the rope arrangement fault using a single frame instead of the comparison of at least two frames, which has lower robustness during high-speed detection;The proposed adaptive Gaussian filtering method ensures the correct extraction of eigenvalues, and the designed distance thresholds of the DBSCAN method enables it to classify eigenvalues earlier and more accurately;It can be adapted to the detection of wide reels in ultra-deep mines; the detection speed of the reel with 95 turns can exceed 200 fps.

This paper presents a high-speed, high-accuracy and high-robust reel rope detection scheme for ultra-deep mines with high-speed and heavy load, which is an important part of ensuring the safety of ultra-deep mines. It is hoped that this projection detection method can provide a new idea for detection in other similar occasions.

## 7. Patents

Tan, J.; Shi, L.; Xue, S.; Wang, Q., A method and device for detecting the rope arranging fault based on projection (P). China. CN201711283787.7. 9 April 2019.

## Figures and Tables

**Figure 1 sensors-21-01769-f001:**
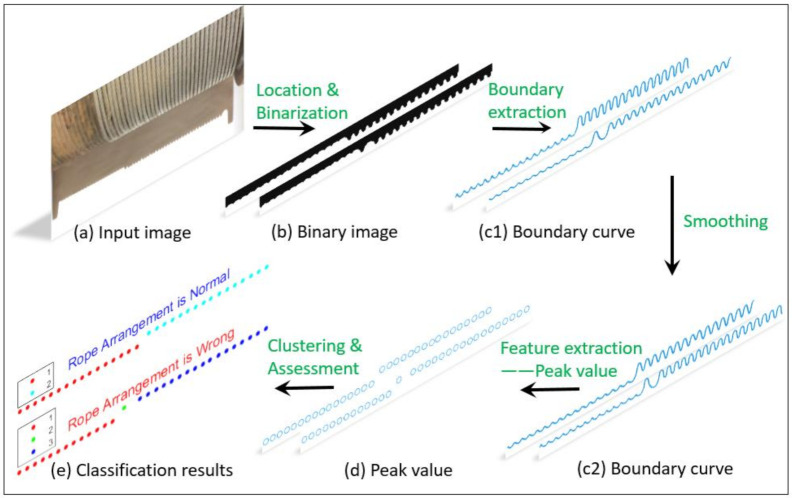
Flowchart of the rope arrangement detection method based on optical projection and machine vision. (**a**) Take a video of the rope arrangement projection, (**b**) the image of projection area in each frame is binarized, (**c1**) extracting edge curve of graph (**b**), (**c2**) smoothing the edge curve of graph (**c1**), (**d**) feature extraction, (**e**) clustering, assessment and obtain the results.

**Figure 2 sensors-21-01769-f002:**
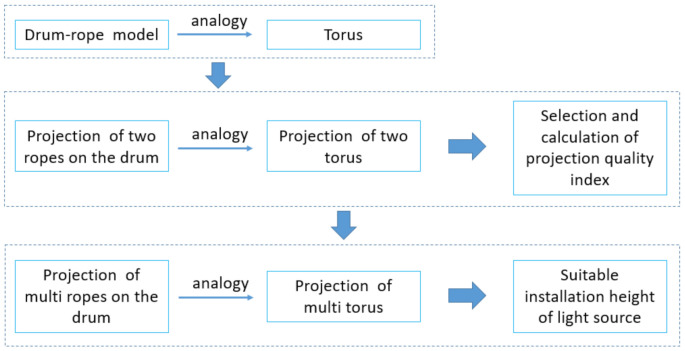
The flowchart of how to get the required projection.

**Figure 3 sensors-21-01769-f003:**
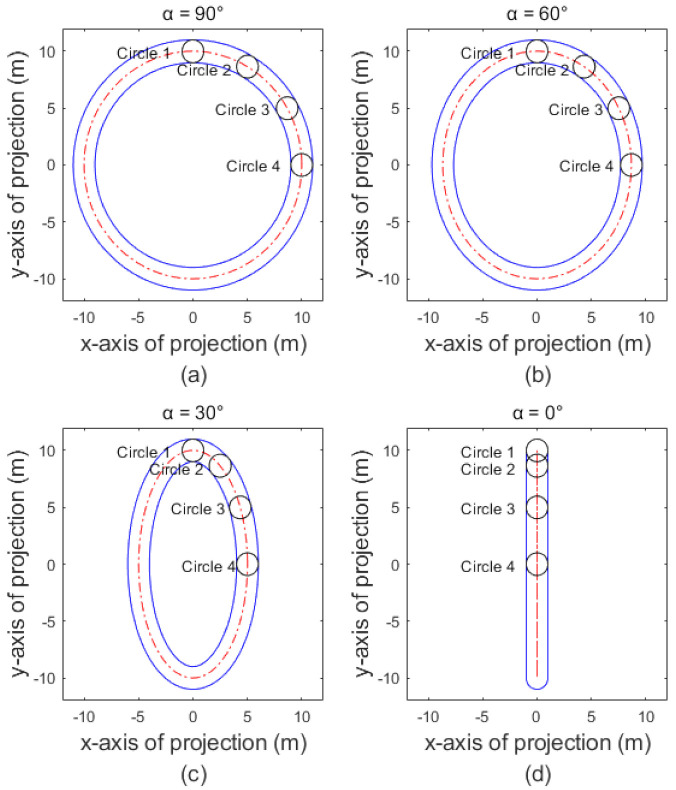
Projection of the single coil wire rope under four projection angles. (**a**) α = 90°; (**b**) α = 60°; (**c**) α = 30°; (**d**) α = 0°. The four small circles in (**a**–**d**) correspond to the projection of a small sphere at the positions of 0°, 30°, 60° and 90°, respectively.

**Figure 4 sensors-21-01769-f004:**
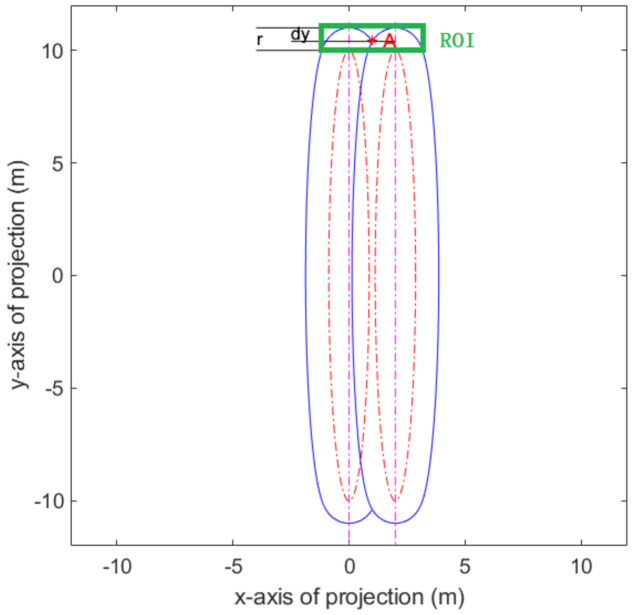
Projection interferogram of two torus. The area of interest (ROI) is shown in the green rectangle. A is the intersection point of the projection outer rings of two torus.

**Figure 5 sensors-21-01769-f005:**
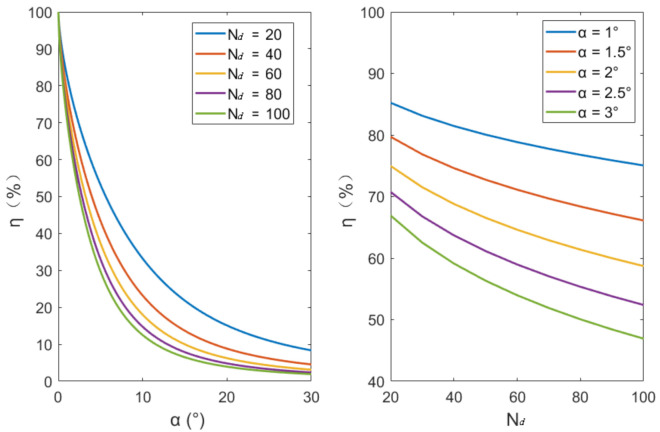
α is the angle between projected light and torus axis, and η is the ratio of height difference of the wire rope projection to the wire rope radius.${\;N}_{d}$ is the ratio of the diameter of the drum to the diameter of the wire rope, usually designed as 40~80.

**Figure 6 sensors-21-01769-f006:**
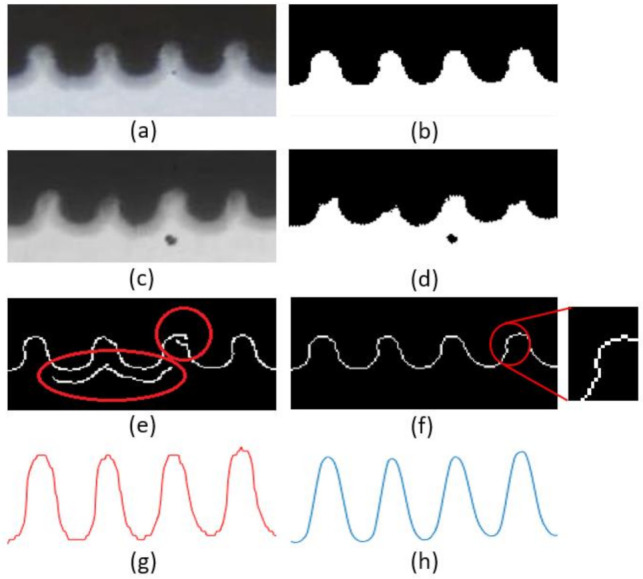
Projection of the drum rope arrangement boundary. (**a**) and (**c**) are the original projection of the projection edge of the drum rope arrangement; (**b**) and (**d**) are, respectively, binary images of (**a**) and (**c**); (**e**) and (**f**) are figures after edge recognition by the Canny algorithm; (**g**) is the contour curve obtained by the method proposed in this study and (**h**) is the result of graph (**g**) after smoothing filtering.

**Figure 7 sensors-21-01769-f007:**
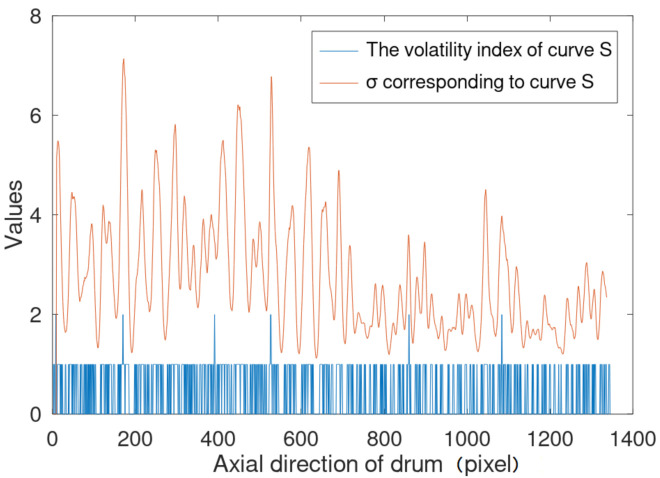
The volatility index of curve S and σ corresponding to curve S of frame 1.

**Figure 8 sensors-21-01769-f008:**
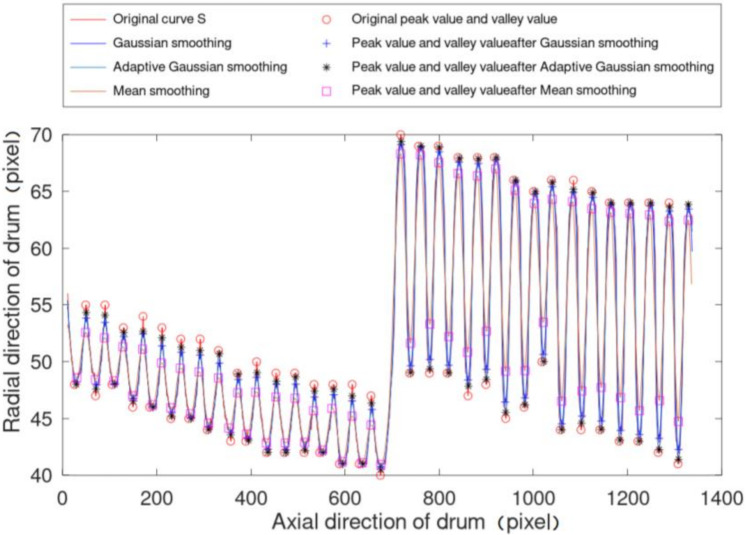
Peak value and valley value of curve S after filtering by three methods in frame 1. The original peak value and valley value are manually annotated.

**Figure 9 sensors-21-01769-f009:**
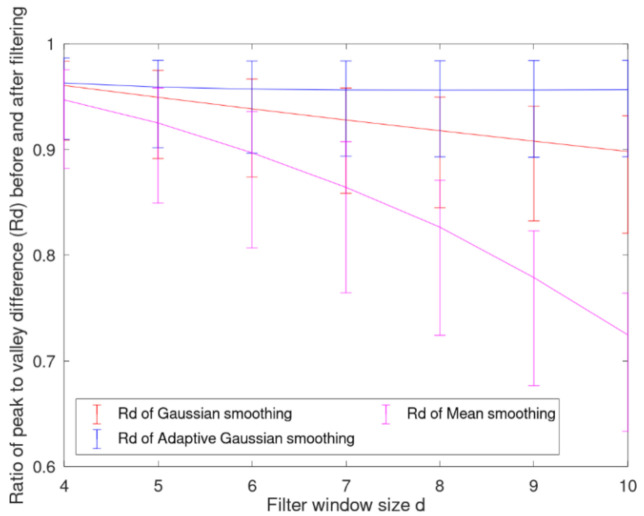
The retention of peak valley difference by the three filtering methods with different window sizes. The upper limit is the average retention rate of the peak valley difference at the projection of the wire rope, and the lower limit is the average retention rate of the peak valley difference at the projection of the rope groove.

**Figure 10 sensors-21-01769-f010:**
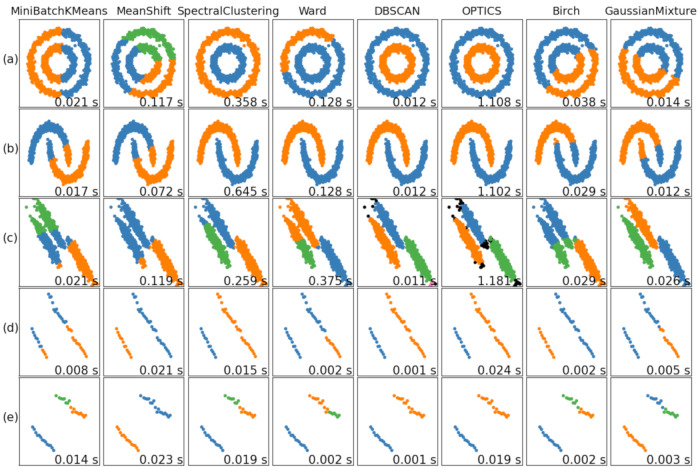
Classification effect of five data sets under eight common clustering classification methods. Datasets: (**a**) The noisy-circles dataset; (**b**) the noisy-moons dataset; (**c**) the aniso dataset; (**d**) the normal rope projection dataset; (**e**) the abnormal rope projection dataset.

**Figure 11 sensors-21-01769-f011:**
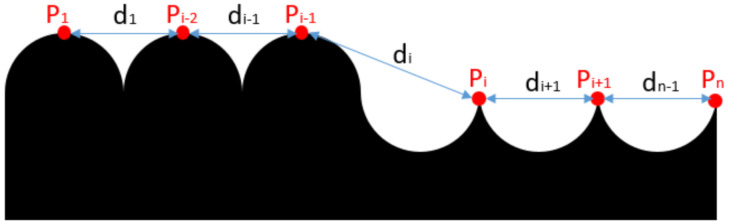
Projection diagram of rope arrangement without rope arrangement fault. “di” is different from the others, while the average of the others is 2 × r.

**Figure 12 sensors-21-01769-f012:**
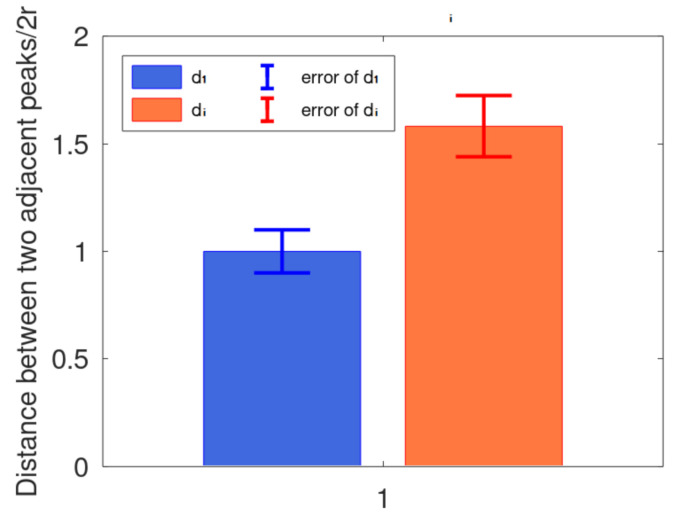
Distance between two adjacent peaks. The mean distance between the two adjacent peaks is 2 × r as d_1_.

**Figure 13 sensors-21-01769-f013:**
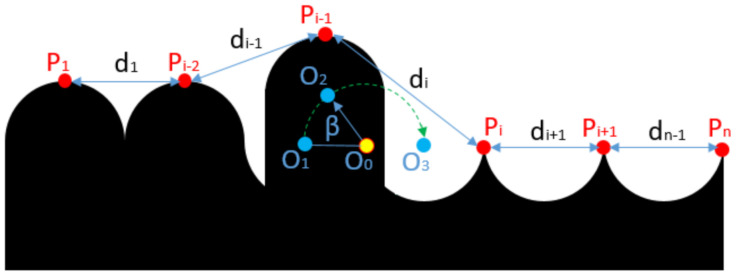
Projection diagram of the rope arrangement in the case of the rope skipping fault.

**Figure 14 sensors-21-01769-f014:**
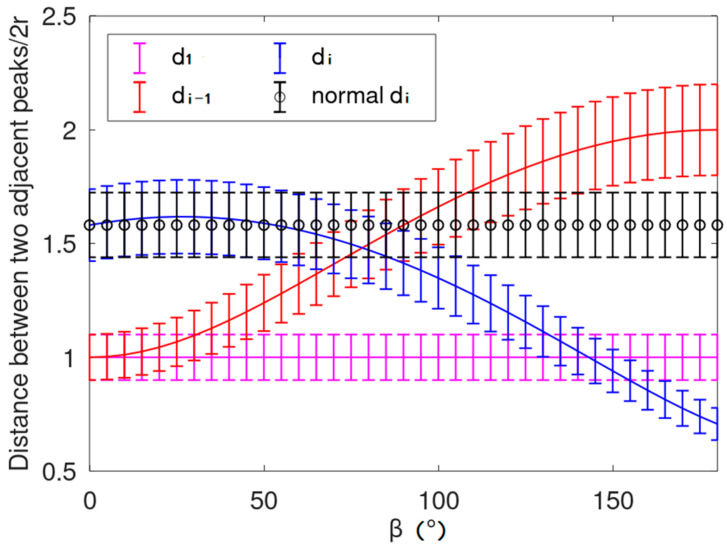
Distance between the two adjacent peaks. The mean distance between the two adjacent peaks is 2 × r as d_1_.

**Figure 15 sensors-21-01769-f015:**
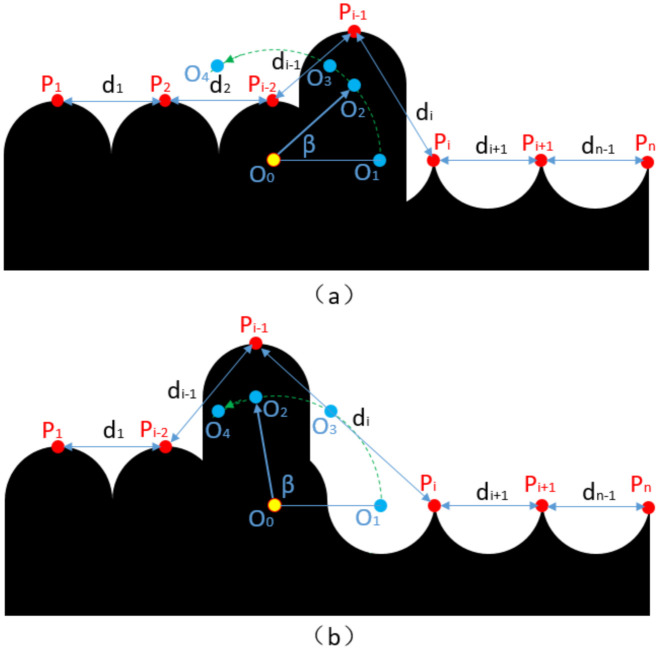
Projection diagram of the rope arrangement in the case of the rope clamping failure. (**a**) β increases from 0° to 60°. (**b**) β increases from 60° to 120°.

**Figure 16 sensors-21-01769-f016:**
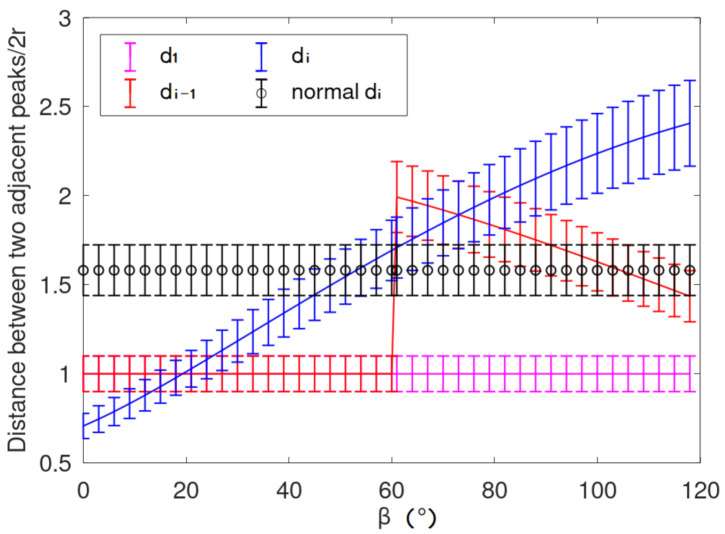
Distance between the two adjacent peaks. The mean distance between the two adjacent peaks is 2 × r as d_1_.

**Figure 17 sensors-21-01769-f017:**
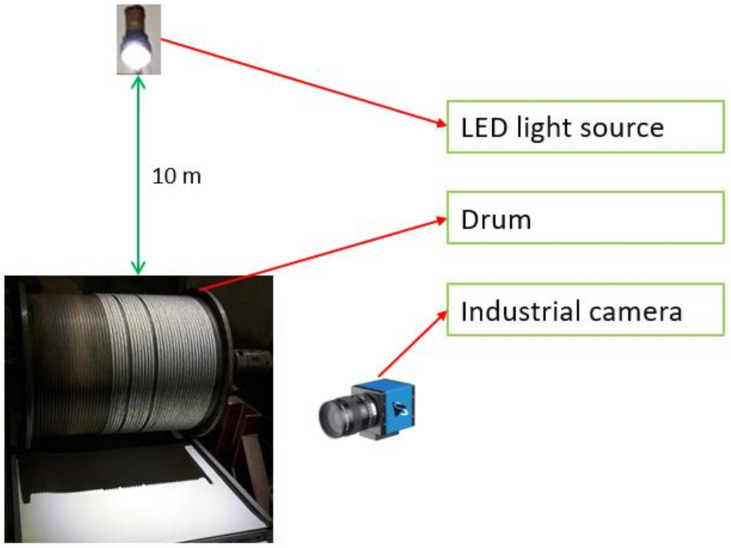
Hardware setting diagram of the laboratory reel rope guiding detection.

**Figure 18 sensors-21-01769-f018:**
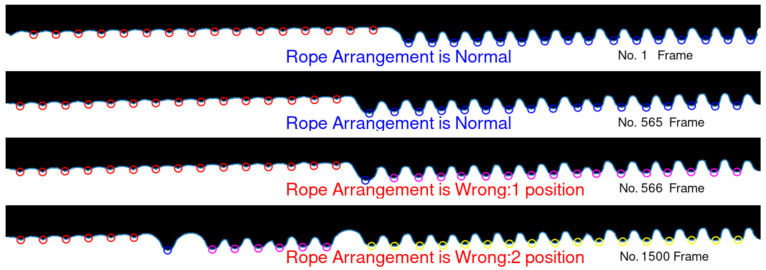
The prediction results before and after the fault in video 4.

**Figure 19 sensors-21-01769-f019:**
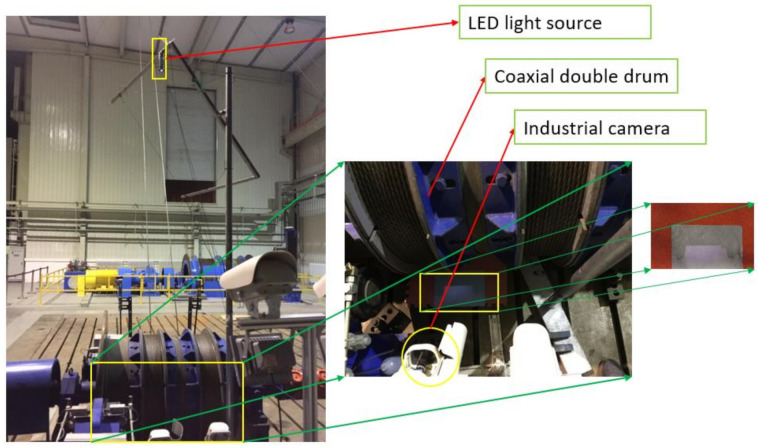
Hardware setting diagram of the drum rope guiding detection for the ultra-deep well simulation test bench.

**Table 1 sensors-21-01769-t001:** Existing detection methods for rope arrangement detection.

Methods	Timeliness	Implementation Difficulty and Cost	Accuracy of Detection	Efficiency of Detection	Robustness
Manual inspections or manual guards	Low (the fault has occurred)	High	Moderate	Very slow	High
Image segmentation [15]	Low (the fault has occurred)	Low	80–90%	100 fps	Low
Template matching [16]	High (happening, at least two frames)	Low	85–90%	30–50 fps	Low
Video tracking (KCF [17])	High (happening, at least two frames)	Low	85–90%	150–200 fps	Low
Video tracking (Staple [18])	High (happening, at least two frames)	Low	90–95%	40–60 fps	Moderate
ours	High (happening, just one frame)	Low	>95%	150–300 fps	High

**Table 2 sensors-21-01769-t002:** Symbols and abbreviations used in this paper.

Symbols and Abbreviations	Meaning	Symbols and Abbreviations	Meaning	Symbols and Abbreviations	Meaning
DBSCAN	Density-based spatial clustering of applications with noise	sum()	A function to calculate the sum	dy	The height difference of the projection intersection point and the highest projection point
KCFs	Kernelized correlation filters	min()	Calculate minimum	η	The ratio between dy and r
ROI	The area of interest	find(equation)	A function to find elements match the requirement	N*_d_*	The ratio between R and r
OPTICS	Ordering points to identify the clustering structure	(x,y), (x_1_,y_1_) (x_2_,y_2_), (x*_t_*,y*_t_*)	Coordinate points	L*_g_*	The width of the drum
G(x)	Gaussian filtering function	R	The radius of the central circle of the torus or the drum	N*_g_*	The ratio between $L_{g}$ and 2 × r
dist()	A function to calculate the distance	r	The radius of the section circle of torus or the radius of the rope	$H_{light}$	The installation height of the point light source
diff()	A function of differential algorithm	$S_{C},S_{C1},S_{C2}$	The central circle projection of the torus	P	Points of eigenvalues
sign()	A function to extract symbols	$S_{Y}$	The external projection of the ring	d	Distance between points of eigenvalues
abs()	A function to calculate the absolute value	α	The angle between the projection light and the plane of the central circle of the torus	β	The skip angle

**Table 3 sensors-21-01769-t003:** The values of μ for commonly used $N_{d}$ and η.

η (%)	$N_{d}=40$	$N_{d}=60$	$N_{d}=80$
50	7.20	8.81	10.19
60	10.68	12.96	15.00
70	16.95	20.91	24.07
80	31.83	39.24	45.47

**Table 4 sensors-21-01769-t004:** The specific parameters of experiments in the laboratory.

Drum’s Width	Drum’s Diameter	Light Source’s Distance	Rope’s Diameter	Number of Rope Winding	$N_{d}$	η	Camera Resolution	Camera Frame Rate
0.5 m	0.3 m	10 m	0.005 m	95	60	70%	1920 × 1680	60 fps

**Table 5 sensors-21-01769-t005:** The specific parameters of the videos and the results of detection.

Videos	Total Frames	Image Size (Pixel × Pixel)	Row Rope Wrong?	Error Sequences	Error Sequence Prediction	Detection Time (s)	Detection Speed (fps)
Video 1	2700	730 × 50	No	\	None	11.8	228
Video 2	2399	1700 × 211	No	\	None	25.5	94
Video 3	1199	1347 × 110	Yes	1–857	1–857	7.8	153
Video 4	2159	1347 × 110	Yes	(1) 567–1427 (2) 1429–2159	(1) 565–1427 (2) 1427–2159	14.1	153

**Table 6 sensors-21-01769-t006:** The specific parameters of experiments in the ultra-deep well simulation test bench.

Drum’s Width	Drum’s Diameter	Light Source’s Distance	Rope’s Diameter	Number of Rope Winding	$N_{d}$	η	Camera Resolution	Camera Frame Rate
0.15 m	0.8 m	3.5 m	0.01 m	14	80	70%	1920 × 1680	60 fps

**Table 7 sensors-21-01769-t007:** The specific parameters of the videos and the results of detection.

Videos	Total Frames	Image Size (Pixel × Pixel)	Row Rope Wrong?	Error Sequences	Error Sequence Prediction	Detection Time (s)	Detection Speed (fps)
Video 5	3565	470 × 70	No	\	None	11.5	310
Video 6	5585	462 × 70	Yes	1128–3423	1126–3423	17.1	326

## Data Availability

Publicly available datasets were analyzed in this study. This data can be found here: (https://pan.baidu.com/s/1zWsrLIYpNIFX2vfI6RZuQw, accessed on 4 March 2021. Extraction code: fuaa).

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
