# Peer review of "Inspection Method of Rope Arrangement in the Ultra-Deep Mine Hoist Based on Optical Projection and Machine Vision"

_sensors, 2021, doi:10.3390/s21051769_

Round 1
Reviewer 1 Report
Article is interesting, the subject is current and has useful value. The citations and references are valid and relevant to the text and bibliographic research and references appear complete and satisfactory. In order to enhance the article quality, I suggest the following remarks be taken into account:
- Line 28: “Deep resource extraction is an effective method for today's energy crisis, and it is also 27 an important strategic goal of the country” What country?
- Please provide the meaning of notations in Equation 1.
- Section 2: The authors should add flowchart of proposed analysis.
- Equation 8: img()?
- Section 3: I suggest that for a better understanding of the paper content and for an easier implementation of the proposed algorithm it would be necessary to rewrite the section by including a flowchart of the algorithm and its algorithmic presentation with all the steps that need to be taken.
Author Response
Thank you for your review. And please see the attachment.

Reviewer 2 Report
The topic is interesting. The paper aims at detecting and diagnosing the problems of rope arranging faults in ultra-deep mines with wide drums and high speeds. The authors proposed a new method for arranging rope detection based on machine vision and optical projection. However, the authors should be taken into account the following aspects to improve the structure and quality of the paper:
- In Section 1, the authors present various approaches from the literature. I think that a synthesis of the solution proposed in the literature depending on the type of analysis is useful for readers. This synthesis can be given as a table.
- The reference loops should be avoided (for example [5] – [7], [10] – [12]) and to be explained what aspects are treated in each reference.
- A list with notations and abbreviations should be introduced.
- The measurement units must be given for all variables and parameters (including in tables and on the coordinate axes of the figures).
- The authors used the variable x in Paragraphs 3.2 and 4.1. It is not clear if the signification is the same. In these conditions, the authors should use different notations.
- The authors assert “DBSCAN does not need to know the number of categories in advance, which has relatively low complexity and fast operation”. Explain better the use of the DBSCAN method because other clustering methods have similar advantages.
- Section 5 should be presented a bit better providing more details to a better understanding.
Author Response

(The authors gave the same response as above.)

Reviewer 3 Report
In this paper, the author developed a machine vision detection method based on the projection of the drum's edge. The novelty and contribution look fine and the subject matter is of interest. The problem is well defined and the results seem compatible. The similarity is fine. I feel no doubt in recommending the paper for publication in the journal.
Author Response
Thank you for your review. It's a great honor to have your affirmation.
Round 2
Reviewer 2 Report
The authors have tried and largely succeeded in removing any doubts raised by the reviewer. They performed changes in the initial manuscript inserting new explanations, elaborations of details, and revisions.